# Decomposition and Carbon and Nitrogen Releases of Twig and Leaf Litter Were Inhibited by Increased Level of Nitrogen Deposition in a Subtropical Evergreen Broad-Leaved Forest in Southwest China

**Yali Song** [1,2] **, Jinmei Xing** [1] **, Chun Hu** [3,*] **, Chenggong Song** [1] **, Qian Wang** [1] **and Shaojun Wang** [1,2,*]

1 College of Ecology and Environment, Southwest Forestry University, Kunming 650224, China; songyali@swfu.edu.cn (Y.S.); xingjinmei1720@163.com (J.X.); m18032147054@163.com (C.S.); qeanqian@163.com (Q.W.)
2 Yuxi Forestry Ecosystem Research Station, National Forestry and Grassland Administration, Kunming 650224, China
3 Department of International Exchange, Joongbu University, Goyang-si 10279, Republic of Korea
* Correspondence: huchun0329@163.com (C.H.); shaojunwang2009@163.com (S.W.)

**Abstract:** Atmospheric nitrogen (N) deposition has rapidly increased due to anthropogenic activities, which can exert a crucial effect on biochemical cycling process such as litter decomposition in the subtropical forests. However, the is still uncertainty about the knowledge of N deposition in regulating nutrient release from the leaf and twig litter. For this study, a 2 yr litterbag decomposition experiment was conducted under three levels of N addition treatments in a subtropical evergreen broad-leaved forest, in southwest China. This study aimed to identify the effects of low (LN: 10 g·N·m$^{-2}$·y$^{-1}$), medium (MN: 20 g·N·m$^{-2}$·y$^{-1}$), and high N addition (HN: 25 g·N·m$^{-2}$·y$^{-1}$) on litter decomposition and nutrient release from leaves and twigs. We observed that there was significantly lower litter decomposition (8.13%–13.86%) and nutrient release (7.24%–36.08%) in the HN treatment compared to the LN treatment. The decay of mass, lignin, and cellulose and the nutrient release were faster in leaf litter than in twig litter after N addition ($p < 0.05$). The ratios of C/phosphorus (P), C/N, and N/P were also significantly greater in twig litter than in leaf litter. Furthermore, the N addition treatments resulted in higher contents of the mass, lignin, and cellulgapose remaining in leaf and twig litter compared to the control (CK). The amount of C, N, and P remaining in leaf (51.4%–59.1%) and twig (44.1%–64.8%) debris was significantly higher in the N treatment compared to CK treatment ($p < 0.05$). In addition, the litter C/N and C/P were smaller and the litter N/P was larger for each N treatment compared to CK ($p < 0.05$). The results suggest that N inputs restrain lignin and cellulose degradation and C and N release, and increase the N/P ratio that limits P release in litter. These effects vary with the level of N treatments.

**Keywords:** atmospheric deposition; decomposition coefficients; carbon release; nitrogen release; phosphorus release; subtropical forest

## 1. Introduction

The deposition of nitrogen (N) has been enhancing rapidly in recent years as a consequence of the massive increase in fossil fuel burning and the increase in agricultural production since the onset of the industrial revolution [1]. It has become a critical contributing factor to climate change at the global scale [2,3]. The estimated global N input is expected to increase from the previous levels of 34 Tg N·y$^{-1}$ in the year 1860 and 123 Tg N y$^{-1}$ in the year 2010 [4] to 200 Tg N·y$^{-1}$ in the year 2050 [5]. In the last 30 years, N deposition in China has accounted for greater than 15% of the global atmospheric N deposition values (increasing 0.041 g·m$^{-2}$·y$^{-1}$) [6,7], with a higher average value than the United States of America and Europe. Therefore, China has become the third largest global

hotspot for N deposition [6]. The average values of N deposition in the subtropical forests of southwest China are 2 to 3 times higher than the national average [6]. High-level and continuous increase in atmospheric N deposition may severely influence the return rate of litter nutrients to the soil [8], soil properties, and the structure of microbial communities and enzyme activity [9], which exerts critical effects on carbon (C), N, phosphorus (P) cycles and nutrients supply in the subtropical forests of southwest China.

The nutrient release from litter is a critical factor in the forest nutrient cycle because the litterfall is returned to the soil through the decomposition accounting for about 50% of the net primary plant production [10,11]. Litter nutrient release is dependent on a variety of biological factors (e.g., litter quality, soil microorganisms, and enzyme activity) and abiotic factors (e.g., climate and chemical composition of litter) [12–14]. Globally, the knowledge about the precise function of factors impacting the litter decomposition varies among different forest ecosystems. Furthermore, several litter decomposition experiments suggest promotion, inhibition, or no impact of N deposition. Jing et al. [15] and Hobbie et al. [16] showed that the N application in *Pinus tabulaeformis* and *Quercus ellipsoidalis* litters significantly accelerated the decomposition and nutrient release due to increased N availability and higher enzyme activities and microbial community in soils. Tie et al. [3] discovered that a lower C/N ratio and litter pH in the soil was associated with the N additions, which inhibited the degradation of litter and nutrient availability in a subtropical forest. However, Wang et al. [17] found no significant difference in litter composition, possibly because of the fact that the stimulatory impact of N addition on the Chinese pine litter may counteract the inhibitory action on the Mongolian oak litter in a temperate forest. The above-mentioned studies primarily discussed the effect of N additions on the decomposition of leaf litter. However, the research work focusing on how twig litter C, N, and P respond to N additions in subtropical forests is relatively limited. Being the second largest component of total litter (26.1%) [18], twig litter plays an important role in the cycling of nutrients in most forest ecosystems. Therefore, it is crucial to assess the responses of N deposition of different litter components (leaves and twigs) simultaneously using the N manipulation experiment in an ecosystem. The findings would further help to understand the adaptation mechanism of forest litter decomposition to nitrogen deposition, especially in subtropical forest ecosystems with high N deposition.

The dry and wet zone in two seasons in southwest China receives considerable N depositions (15 and 10 g N·m$^{-2}$·yr$^{-1}$, respectively) [19,20], values which are remarkably greater than the mean value across China. In addition, the extremely high N input levels constrain the processes of decomposition of litter and the release of nutrients in the forest area. It also represents a significant region of acid deposition in China [21]. Nevertheless, N deposition might have different effects on degradation and nutrient mineralization from component to litter component, due to different the area where different components interact with soil animals and microorganisms and the content of refractory lignin. This study involved a two-year experiment conducted in a field in southwest China's subtropical evergreen broad-leaved forest, where the region has obvious climatic characteristics of dry and wet two seasons. The effects of simulated N deposition on the decomposition of leaf and twig litter and C, N, and P release were studied. Previous studies conducted in different forests during this two-season area have revealed that the N addition decreased the degradation rate of cellulose, lignin [3], soil microbial diversity, network complexity [22,23], and soil C-cycle enzyme activities [24]. This research aims to enhance the comprehensive understanding of the impacts of N deposition on the nutrients release of different components in litter in subtropical evergreen broad-leaved forests of this work's study area. It also aims to provide a theoretical basis for predicting the response and adaptation of the forest litter components decomposition process to continuous nitrogen increase in the high-altitude region. It suggests that the input of N might negatively impact the decomposition of litter and the release of nutrients [25,26]. We speculated that (1) the N additions would decrease the rate of decay and the release of C, N, and P nutrients from the two types of litters via changing lignin and cellulose degradation (i.e., decreasing lignin and cellulose

degradation can slow down the release of litter nutrients); and (2) the effect of N additions on litter decomposition and nutrient release would vary between two litter types due to the conspicuous differences in initial physical properties and stoichiometric ratios in leaf and twig litters.

## 2. Materials and Methods

### 2.1. Study Area

An evergreen broad-leaved forest selected in the current study is located in the Mopan Mountain (23°46′18″–23°54′34″ N, 101°16′06″–101°16′12″ E, 2270 m a.s.l.), National Positioning observation and research station in Yuxi Forest Ecosystem, Southwestern China. The mean annual temperature (MAT) and mean annual precipitation (MAP) of the last decade were 15.1 °C and 1050 mm, respectively. Mopan Mountain belongs to the subtropical inland plateau climate, with large elevation differences and obvious vertical climate change. Climatic conditions vary from the south subtropical climate at the bottom of the mountain gully to the northern subtropical climate at the top. In addition, the alpine meadow in the middle of the mountain belongs to the middle subtropical climate. This area comprises mainly primary and secondary original forest areas with a forest coverage rate of 86%. The tree species dominant in this subtropical evergreen broad-leaved forest are *Castanopsis carlesii* (60%), *Lithocarpus mairei*, *Betula utilis*, *Rhododendron delavayi*, and *Dichotomanthes tristaniicarpa*, with a standard of approximately 20 years (Table 1). The soil mainly consists of ArgiUdic Ferrosols (Soil Taxonomy data from the United States Department of Agriculture). The soil layer was about 60–80 cm deep in the study area. Basic information of soil properties are shown in Table 2.

**Table 1.** Characteristics of the sample plots in subtropical evergreen broad-leaved forest.

| Stand | Altitude/m | Age/a | Mean Height H/m | DBH/cm | Canopy Density | Slope/(°) | Aspect | Soil Category |
|---|---|---|---|---|---|---|---|---|
| 1 | 2130 | 20 | 9.6 | 14.5 | 0.87 | 23 | NE | red soil |
| 2 | 2132 | 18 | 12.1 | 20.7 | 0.9 | 28 | NE | red soil |
| 3 | 2133 | 20 | 10.8 | 18.3 | 0.85 | 30 | NE | red soil |

**Table 2.** Basic information of soil properties.

| Treatment | Soil Layer (cm) | Maximum Water-Holding Capacity (mm) | Capillary Water Capacity (mm) | Field Water Capacity (mm) | Overall Porosity (%) | Capillary Porosity (%) | Noncapillary Porosity (%) | Moisture Content (%) | Bulk Density (g·cm$^{-3}$) | Organic Carbon (mg·g$^{-1}$) | Total Nitrogen (mg·g$^{-1}$) | Total Phosphorus (mg·g$^{-1}$) |
|---|---|---|---|---|---|---|---|---|---|---|---|---|
| CK | 0~10 | 62.07 ± 3.22 b | 54.28 ± 1.44 b | 44.88 ± 0.17 b | 53.03 ± 1.73 a | 55.32 ± 7.84 a | 10.23 ± 2.53 a | 47.87 ± 1.80 a | 0.95 ± 0.10 a | 40.29 ± 1.23 a | 0.90 ± 0.05 a | 0.66 ± 0.01 a |
| | 10~20 | 103.27 ± 12.75 a | 93.41 ± 3.85 a | 70.91 ± 6.51 a | 44.33 ± 3.38 b | 49.38 ± 6.98 b | 8.90 ± 5.13 b | 34.38 ± 4.69 b | 1.15 ± 0.08 a | 21.29 ± 0.54 b | 0.43 ± 0.01 b | 0.42 ± 0.02 b |
| LN | 0~10 | 53.03 ± 1.72 b | 42.80 ± 4.25 b | 33.59 ± 1.60 b | 65.21 ± 1.22 a | 54.28 ± 1.44 a | 6.72 ± 4.62 a | 43.13 ± 1.14 a | 0.93 ± 0.11 a | 40.14 ± 1.87 a | 0.89 ± 0.02 a | 0.64 ± 0.03 a |
| | 10~20 | 96.67 ± 6.77 a | 78.88 ± 3.48 a | 63.67 ± 1.72 a | 52.06 ± 0.33 b | 46.71 ± 1.93 b | 2.25 ± 0.61 b | 28.32 ± 1.09 b | 0.95 ± 0.11 a | 20.22 ± 0.36 b | 0.41 ± 0.01 b | 0.41 ± 0.01 b |
| MN | 0~10 | 65.21 ± 1.22 b | 49.96 ± 4.84 b | 45.30 ± 5.53 b | 61.72 ± 11.98 a | 49.96 ± 4.48 a | 10.93 ± 2.65 a | 49.51 ± 2.02 a | 0.88 ± 0.09 b | 38.46 ± 0.98 a | 0.88 ± 0.04 a | 0.62 ± 0.05 a |
| | 10~20 | 104.12 ± 0.66 a | 103.05 ± 12.58 a | 81.78 ± 8.52 a | 56.11 ± 2.82 b | 51.52 ± 6.29 a | 5.35 ± 2.26 b | 35.60 ± 0.66 b | 1.27 ± 0.14 a | 21.65 ± 0.13 b | 0.37 ± 0.02 b | 0.38 ± 0.02 b |
| HN | 0~10 | 61.72 ± 11.98 b | 55.32 ± 7.84 b | 41.04 ± 2.91 b | 62.07 ± 3.22 a | 42.80 ± 4.25 a | 11.76 ± 7.14 a | 47.34 ± 7.20 a | 1.10 ± 0.11 a | 39.46 ± 1.54 a | 0.88 ± 0.04 a | 0.69 ± 0.04 a |
| | 10~20 | 112.22 ± 5.64 a | 98.77 ± 13.97 a | 87.67 ± 16.62 a | 51.63 ± 6.38 b | 39.44 ± 1.74 b | 4.59 ± 3.47 b | 48.12 ± 11.80 a | 1.23 ± 0.10 a | 19.03 ± 0.11 b | 0.42 ± 0.01 b | 0.40 ± 0.01 b |

Different letters of data in the same column indicated significant differences in indexes of different soil layers ($p < 0.05$).

## 2.2. Design of Experimentation

### 2.2.1. The Collection and Bagging of Leaf and Twig Litters

Freshly fallen leaf and twig litter were collected from the ground consecutively for the two-year litter decomposition experiment. The litter collected from the study area was air-dried in the laboratory. In the current study, we used litter that was mixed to simulate the litter decomposition process in a natural setting. The air-dried litter proportion of *Castanopsis carlesii* to other plants were 6:4. We weighed ten bags of each component of the air-dried litter randomly and placed it into an oven at 65 °C for a period of 96 h before evaluating the initial chemical properties. Then, a 10.0 g ($\pm$0.01 g) of mixed air-dried litter was placed into nylon mesh litter bags (20 $\times$ 20 cm, 1.0 mm on the surface pore size, and 0.05 mm on the bottom). A total of 576 litter bags (4 treatments $\times$ 2 components $\times$ 3 replicates $\times$ 12 months/year $\times$ 2 years) were filled and positioned on the surface in the particular treatment section in Jan 2019. The initial concentrations of different components in leaf and twig litters (average $\pm$ standard deviation) were measured [27] as follows: C 501.68 $\pm$ 2.70 and 541.22 $\pm$ 5.18, N 16.21 $\pm$ 0.32 and 8.32 $\pm$ 0.18, P 1.76 $\pm$ 0.05 and 0.88 $\pm$ 0.03, lignin 395.22 $\pm$ 2.29 and 399.37 $\pm$ 3.40, and cellulose 205.71 $\pm$ 3.47 and 306.10 $\pm$ 2.85 mg g$^{-1}$, respectively.

### 2.2.2. Design of Plot and Fertilization

According to the previous literature, high levels of ambient atmospheric wet N deposition, and considering the possibility of future N deposition trends in southwest China [3,21,23,24,28], four N addition levels of 3 $\times$ 3 m$^2$ in triplicates (randomized complete block design, each plot was separated by >10 m) were established for this experiment at the beginning of January 2019. These levels comprised the control (Control: 0 g$\cdot$N$\cdot$m$^{-2}\cdot$y$^{-1}$), low-N addition (LN: 10 g$\cdot$N$\cdot$m$^{-2}\cdot$y$^{-1}$), medium-N addition (MN: 20 g$\cdot$N$\cdot$m$^{-2}\cdot$y$^{-1}$), and high N addition (HN: 25 g$\cdot$N$\cdot$m$^{-2}\cdot$y$^{-1}$). In early Jan 2019, 72 (24 samplings by three bags sampling$^{-1}$ plot$^{-1}$) leaf or twig litter bags were placed evenly on the soil surface of each plot. Four treatments were conducted randomly in different plots. Urea [CO(NH$_2$)$_2$] was used as the nitrogen source for treatment, which was dissolved in 1 L of water. The two-year application amount was applied monthly using a hand-held sprayer (with a maximum capacity of 5 L) in 24 equal applications from January 2019 to December 2019 (the first year) and from June 2020 to May 2021 (the second year due to the impact of COVID-19). The plots of CK were sprayed with 5 L of water (no nitrogen addition). Nitrogen treatment was carried out on the quadrat in the middle of every month according to the above-mentioned levels. There was no substantial difference in the soil moisture content or temperature following control and different treatments of N addition [21].

## 2.3. Sampling and Assessment of Litter

### 2.3.1. Litter Sampling

Litter bags were harvested every month (continuously for 24 months) from each plot from February 2019 after N application. During each sampling time, three litter bags of leaves or twigs were randomly collected from every plot of the study area. Soil particles, roots, and other extraneous materials were taken out from the litter that was harvested, and the duly cleaned samples were dried in the oven for 48 h at 65 °C to constant mass. The litter samples were ground and filtered (<0.15 mm) before assessing the contents of cellulose, lignin, and nutrient elements.

### 2.3.2. Chemical Analysis

For chemical analysis, the filtered samples were used. The contents of organic carbon (OC), total phosphorus (TP), and total nitrogen (TN) in the litter were assessed with the dichromate oxidation–external heating method, molybdenum–antimony colorimetry method, and Kjeldahl digestion method, respectively [27]. The acid unhydrolyzable residue (AUR) of lignin and cellulose contents was evaluated utilizing a digestion method based on acid detergent [29]. All the experiments were conducted repeatedly three times.

*2.4. Determination of the Stoichiometry, Remaining Mass, and Remaining Nutrients in Litter*

The stoichiometric ratios of leaf and twig litters (C:N, C:P, and N:P) were determined according to the element mass.

The remaining mass (RM) was determined according to Equation (1) [21]:

$$R_m\ (\%) = \frac{W_t}{W_0} \times 100\% \tag{1}$$

where $R_m$ represents the RM of the initial amount (%), $W_0$ represents the initial weight of air-dried leaf or twig litter (in grams), and $W_t$ represents the dry weight of litter at the $t$ th sampling time (in grams).

The remaining amounts of lignin, cellulose, C, N, and P in leaf and twig litters were determined by Equation (2) [30]:

$$R_y\ (\%) = \frac{W_t C_t}{W_0 C_0} \times 100\% \tag{2}$$

where $R_y$ represents the remaining amount of the initial amount (%), $C_0$ represents the initial nutrient concentration (in mg/g), and $C_t$ represents the nutrient concentration at the $t$ th sampling time (mg/g).

Decomposition rates of leaf and twig litters with time were evaluated using the dry mass which remains, as per the negative exponential decay model described by Olson [31] (3):

$$y = e^{-kt} \tag{3}$$

where $y$ represents the rate of decomposition of leaf and twig litters at time $t$, $k$ is the yearly decomposition coefficient, and $t$ represents the time elapsed in years.

The time for 50% of 95% decomposition of litter ($T_{50\%}$ and $T_{95\%}$) was determined following Equations (4) and (5) [32]:

$$T_{50\%} = -ln(1 - 0.50)/k \tag{4}$$

$$T_{95\%} = -ln(1 - 0.95)/k \tag{5}$$

*2.5. Statistical Analyses*

Before conducting the statistical analysis, the data were subjected to quality check to assess whether they conformed to a normal distribution and homogeneity of each variable or not. Data that did not follow these pre-conditions were converted into unequal or non-normal variances before analysis. We performed repeated measures analysis of variance (ANOVA) to assess the additional effect in terms of the remaining litter mass, C, N, P, lignin, and cellulose; the stoichiometry ratio of litter at the end of the experiment; the k-value; and the $T_{50\%}$ and $T_{95\%}$. Regression analysis was conducted to analyze the relationships among the remaining C, N, P, lignin, and cellulose. Differences among mean values were considered statistically significant at $p < 0.05$ levels (unless otherwise stated). The Statistical Package for the Social Sciences (SPSS) software package (version 25.0 for Windows) was used to perform all the statistical analyses.

## 3. Results

### 3.1. Litter Remaining and Decomposition Coefficient

We observed a lower remaining mass in leaf litterbags (28.1%) than in twig litterbags (52.1%) through 24-month decomposition (Figure 1). The decomposition rates of two litter types both decreased with the increasing N application rates ($p < 0.05$). The inhibition ratio of leaf litter in LN, MN, and HN was 10.3%, 18.0%, and 30.3%, respectively, whereas that of twig litter was 2.0%, 7.9%, and 14.4%., respectively (Figure 1). These results highlight that leaf litter decomposed faster than twig litter after N addition.

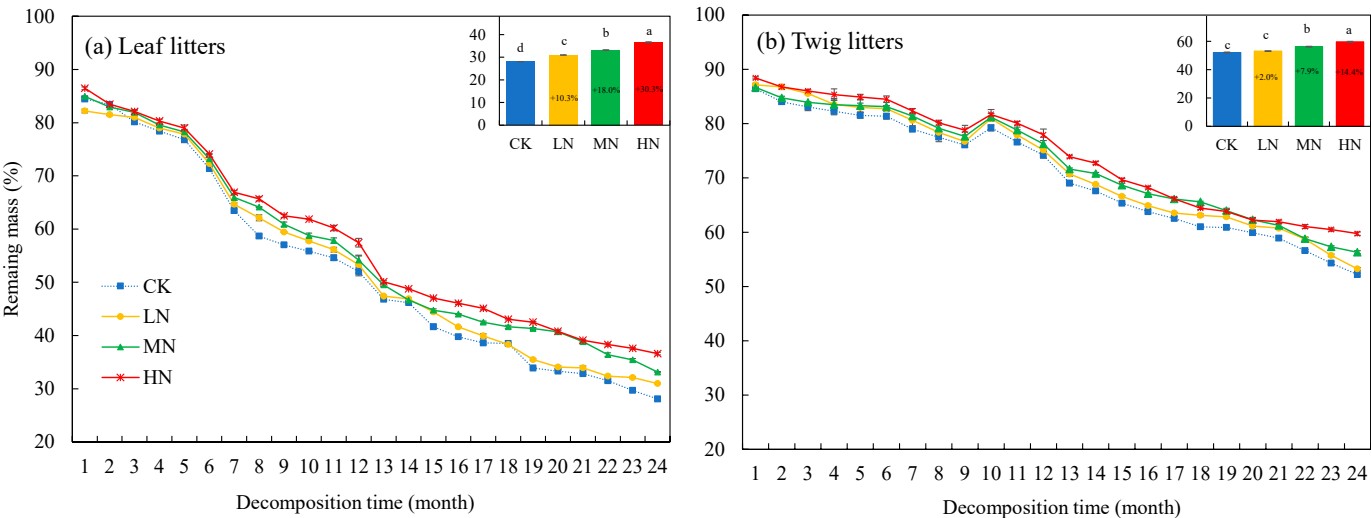

**Figure 1.** Remaining mass (% of the initial mass) in leaf (**a**) and twig (**b**) litters during the decomposition process. Values are mean ± SE (standard error). Data in the upper right figure indicate inhibition ratio: inhibition ratio (%) = (T (treatment) − CK)/CK. Different lowercase letters in the histograms indicate significant differences among four experimental treatments, based on one-way ANOVA ($p < 0.05$). Months 1–12 in the figure represent decomposition time (month) from February 2019 to January 2020, and months 13–24 represent decomposition time (month) from July 2020 to June 2021. The same is true below.

The k-values of two litter types decreased and the $T_{50\%}$ and $T_{95\%}$ increased with the increasing levels of N treatments during the study period (Table 3; $p < 0.05$). The respective $T_{50\%}$ and $T_{95\%}$ values were greater in the LN (1.212 and 5.237 year), MN (1.362 and 5.886 year), and HN (1.429 and 6.177 year) treatments of leaf litter than for the CK treatment (1.2149 and 4.968 years; $p < 0.05$). The respective $T_{50\%}$ and $T_{95\%}$ values were greater in the LN (2.740 and 11.841 year), MN (3.014 and 13.025 year), and HN (3.054 and 13.19 year) treatments of twig litter than for the CK treatment (2.687 and 11.611 years; $p < 0.05$). These results indicated that three levels of N additions decreased the decomposition rates of leaf and twig litter. Furthermore, the effects of litter types, sampling time, N addition level, and their interactions had significant impacts on the remaining mass of leaf and twig litters (Table 4).

**Table 3.** Decomposition coefficient (*k*-value) and time for 50% and 95% decomposition ($T_{50\%}$ and $T_{95\%}$) under various treatments based on regression analyses.

| | Treatments | Regression Equation | Coefficient of Determination ($R^2$) | *k*-Value (Year$^{-1}$) | $T_{50\%}$ (Year) | $T_{95\%}$ (Year) |
|---|---|---|---|---|---|---|
| Leaf litter | CK | $y = 92.421e^{-0.603t}$ | 0.991 ** | −0.603 | 1.149 | 4.968 |
| | LN | $y = 91.982e^{-0.572t}$ | 0.989 ** | −0.572 | 1.212 | 5.237 |
| | MN | $y = 90.605e^{-0.509t}$ | 0.985 ** | −0.509 | 1.362 | 5.886 |
| | HN | $y = 91.212e^{-0.485t}$ | 0.985 ** | −0.485 | 1.429 | 6.177 |
| Twig litter | CK | $y = 91.294e^{-0.258t}$ | 0.965 ** | −0.258 | 2.687 | 11.611 |
| | LN | $y = 92.727e^{-0.253t}$ | 0.965 ** | −0.253 | 2.740 | 11.841 |
| | MN | $y = 91.824e^{-0.230t}$ | 0.960 ** | −0.230 | 3.014 | 13.025 |
| | HN | $y = 93.159e^{-0.227t}$ | 0.964 ** | −0.227 | 3.054 | 13.197 |

Values are the means of three plot replicates ± SE. The abbreviations coined for different treatments are provided in Figure 1. ** Represents high statistical significance ($p < 0.01$). Various superscript lowercase letters represent significant differences between various treatments ($p < 0.05$).

**Table 4.** Repeated-measure ANOVA for the impacts of litter type, components (CP, represents components of leaf and twig litters), sampling time (T), N addition levels (N), and their interactions (T × N, T × CP, N × CP, T × N × CP) on remaining litter, lignin, cellulose, and nutrient elements (CP, N, and P) and their stoichiometric ratios.

| | | Litter Remaining | Lignin Remaining | Cellulose Remaining | C Remaining | N Remaining | P Remaining | C/N | C/P | N/P |
|---|---|---|---|---|---|---|---|---|---|---|
| CP | df | 1 | 1 | 1 | 1 | 1 | 1 | 1 | 1 | 1 |
| | MSE | 44,991.71 | 5501.931 | 55,162.547 | 34,799.348 | 195,217.799 | 287,360.77 | 47,911.008 | 37,269,994.3 | 57.678 |
| | *p* | <0.001 | <0.001 | <0.001 | <0.001 | <0.001 | <0.001 | <0.001 | <0.001 | <0.001 |
| T | df | 23 | 23 | 23 | 23 | 23 | 23 | 23 | 23 | 23 |
| | MSE | 4559.169 | 1454.799 | 3647.243 | 70,007.933 | 8343.27 | 9020.417 | 873.45 | 119,925.359 | 66.652 |
| | *p* | <0.001 | <0.001 | <0.001 | <0.001 | <0.001 | <0.001 | <0.001 | <0.001 | <0.001 |
| N | df | 3 | 3 | 3 | 3 | 3 | 3 | 3 | 3 | 3 |
| | MSE | 499.32 | 2854.963 | 1880.606 | 1287.374 | 20,112.376 | 5873.764 | 2371.048 | 3437.328 | 89.86 |
| | *p* | <0.001 | <0.001 | <0.001 | <0.001 | <0.001 | <0.001 | <0.001 | <0.05 | <0.001 |
| T × N | df | 69 | 69 | 69 | 69 | 69 | 69 | 69 | 69 | 69 |
| | MSE | 4.026 | 12.424 | 6.526 | 10.245 | 258.53 | 444.702 | 55.159 | 5292.109 | 4.694 |
| | *p* | <0.001 | <0.001 | <0.001 | <0.001 | <0.001 | <0.001 | <0.001 | <0.001 | <0.001 |
| T × CP | df | 23 | 23 | 23 | 23 | 23 | 23 | 23 | 23 | 23 |
| | MSE | 394.106 | 171.368 | 628.9436 | 840.931 | 1665.793 | 4456.391 | 714.259 | 58,060.194 | 28.887 |
| | *p* | <0.001 | <0.001 | <0.001 | <0.001 | <0.001 | <0.001 | <0.001 | <0.001 | <0.001 |
| N × CP | df | 3 | 3 | 3 | 3 | 3 | 3 | 3 | 3 | 3 |
| | MSE | 20.727 | 344.764 | 46.641 | 21.921 | 1435.881 | 339.5 | 303.75 | 6463.765 | 4.766 |
| | *p* | <0.01 | <0.001 | <0.001 | <0.001 | <0.001 | <0.001 | <0.001 | <0.001 | <0.001 |
| T × N × CP | df | 69 | 69 | 69 | 69 | 69 | 69 | 69 | 69 | 69 |
| | MSE | 2.675 | 5.621 | 4.699 | 4.821 | 198.608 | 202.976 | 40.425 | 2787.844 | 2.026 |
| | *p* | <0.001 | <0.001 | <0.001 | <0.001 | <0.001 | <0.001 | <0.001 | <0.001 | <0.001 |

*3.2. Lignin and Cellulose Remaining in the Litter*

We observed a lower lignin remaining in leaf litterbags (46.0% ± 0.06%) than in twig litterbags (56.7% ± 0.37%) through 24-month decomposition (Figure 2). The decomposition rates of two litter types both decreased with the increasing N application rates (*p* < 0.05). The lignin remaining content of leaf litter in the LN, We're not sure where is the problem MN, and HN was 5.4%, 23.3%, and 30.4% higher than CK, respectively, while the remaining in twig litter was 0.8%, 6.8%, and 14.5% higher than CK, respectively. The leaf litter had lower cellulose content remaining (29.5% ± 0.04%) than in twig litter (56.6% ± 0.37%), and with the decomposition time, the cellulose content of twig litter gradually decreased. The remaining cellulose content of leaf litter in the LN, MN, and HN was 3.2%, 15.6%, and 32.7%, respectively, while the remaining in twig litter was 6.7%, 8.8%, and 14.9%, respectively.

The results showed that lignin and cellulose in both leaf and twig litters decomposed slowly in the HN treatment compared to LN and MN treatments (*p* < 0.05). Furthermore, the nitrogen addition promotion affected lignin and cellulose decomposition more in leaf litter than in twig litter.

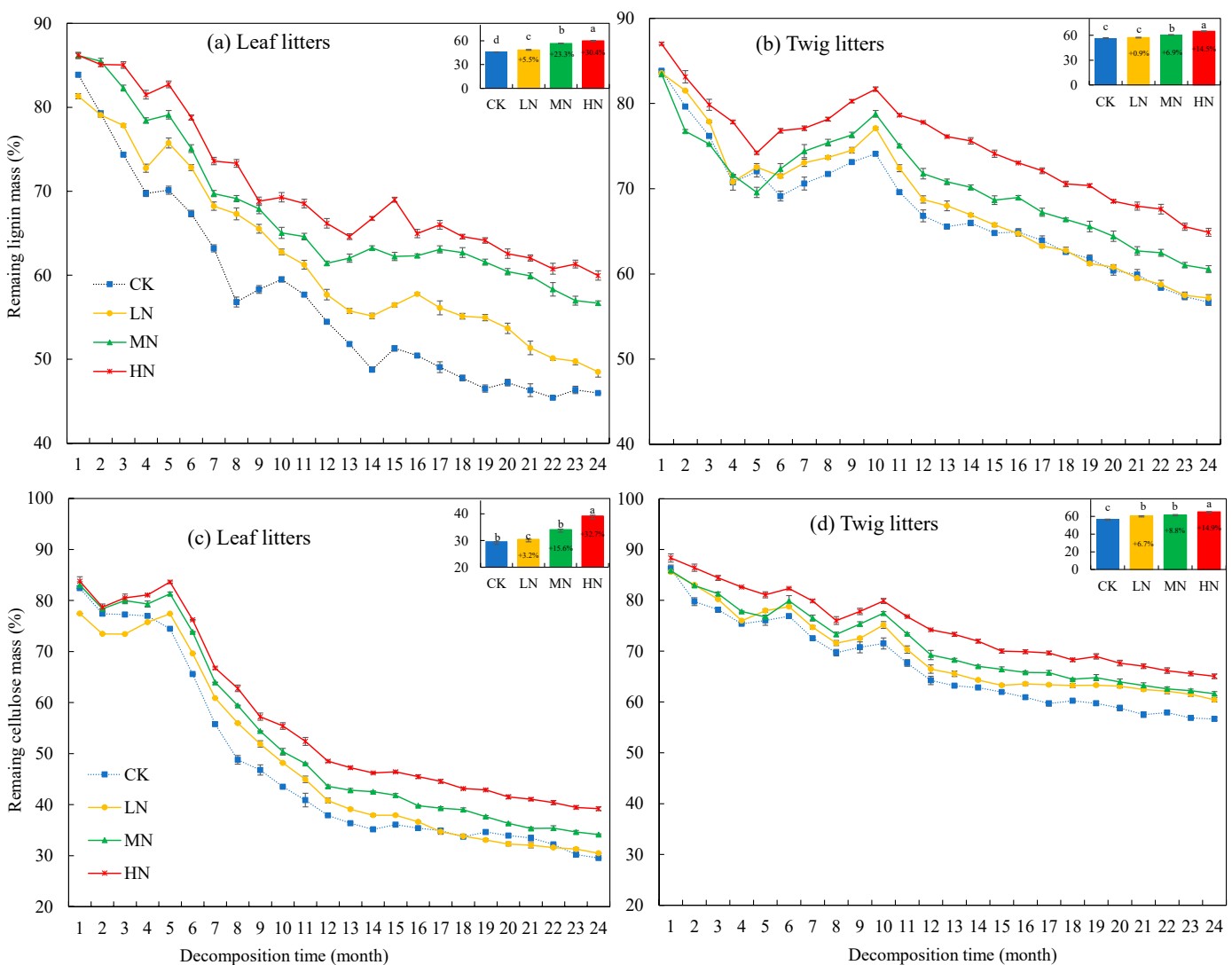

**Figure 2.** Lignin and cellulose mass remaining (% of the initial content) in leaf litters during the decomposition process (**a** or **c**) and twig (**b** or **d**). Estimated values are the average of three plot replicates ± SE (*n* = 3). Various lowercase histogram letters in top right position represent statistically significant differences between various treatments at the end of the experiment based on three-way ANOVA (*p* < 0.05). The abbreviations for different treatments are provided in Figure 1.

*3.3. Nutrients Remaining in the Litter*

At the end of the experiment, the amount of C, N, and P remaining in the N-treated leaf and twig debris was significantly higher than in the CK treatment (*p* < 0.05; Figure 3). C, N, and P in the leaf litter were released rapidly, showed temporal dynamics and remained at 22.6%–31.4%, 18.2%–29.4%, and 41.4%–63.6%, respectively, two years after abscission (Figure 3a,c,e). Compared to CK, they increased by 8.8%–38.8%, 31.0%–61.2%, and 24.4%–53.4%, respectively. Similarly, carbon, nitrogen, and phosphorus residues in the twig litter were 51.4%–59.1% and 44.1%–64.8% by the end of the experiment (Figure 3b,d,f), which were increased by 14.5%, 1.8%–46.8%, and 9.3%–19.2%, respectively, compared to CK.

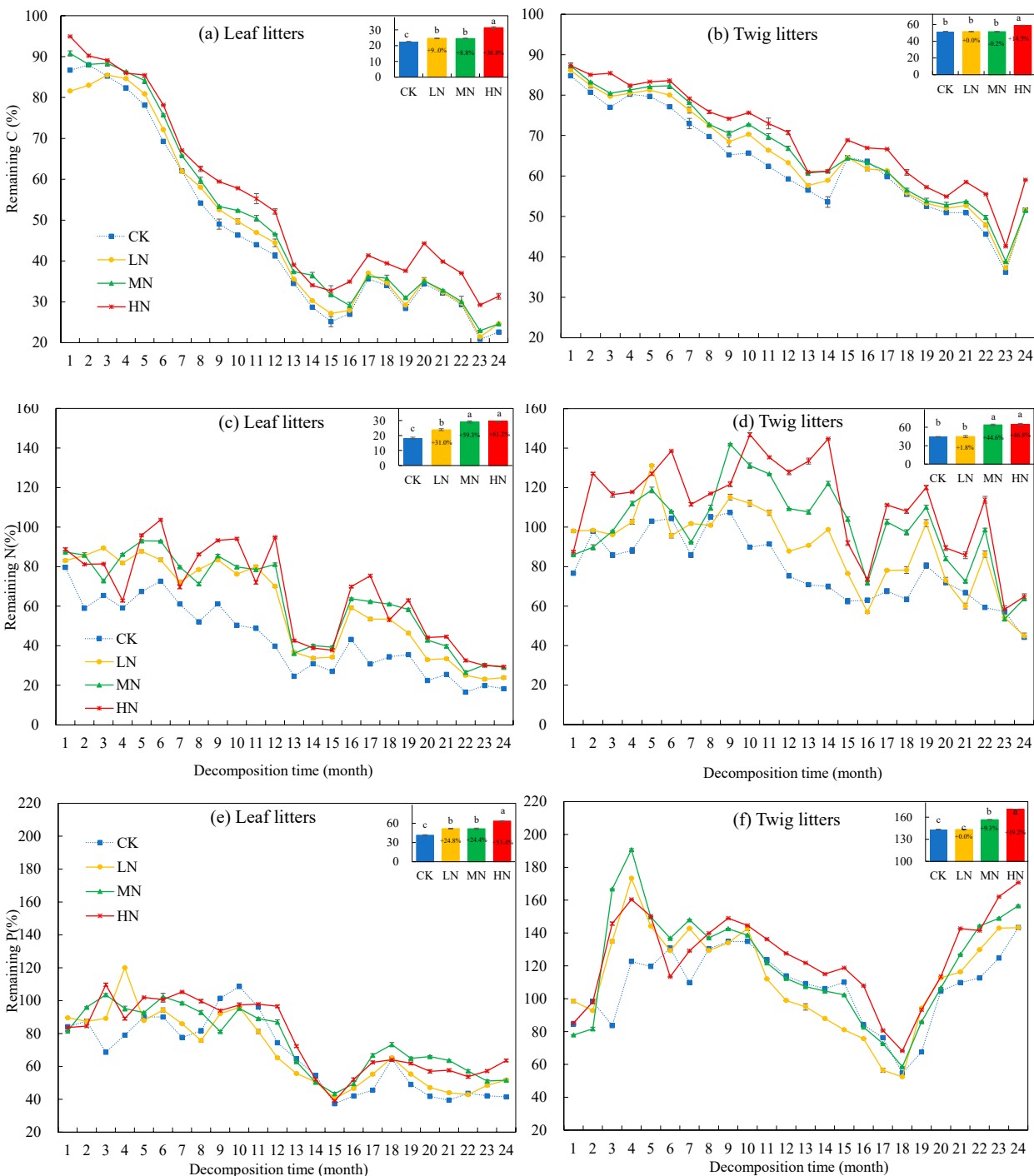

**Figure 3.** Remaining C (**a**,**b**), N (**c**,**d**), and P (**e**,**f**) in leaf and twig litters (% of the initial amount). Different histogram lowercase letters represent statistically significant differences between separate treatments at the end of the experiment on the basis of three-way ANOVA (*p* < 0.05). Estimated values are the average of three plot replicates ± SE (*n* = 3).

Furthermore, they were significantly higher in the HN treatment compared to the LN treatment (*p* < 0.05). There was no significant difference in the N content in the remaining litters between MN and HN treatments. The effects of sampling time, litter type, and N addition level on the remaining C, N, and P (Table 2, *p* < 0.05). Hence, we speculated that these additional treatments inhibited the release of C, N, and P from both leaf and twig litter, and leaf litter had a higher release as compared to twig litter after the N addition.

### 3.4. Stoichiometric Ratios in the Litter

The leaf C/N, C/P, and N/P stoichiometric ratios were 30.82–38.24, 140.80–159.53, and 4.04–5.18, respectively, for different decomposition processes, whereas for twigs litter they were 56.60–76.19, 205.65–221.01, and 2.81–3.77, respectively (Figure 4). The ratios of litter C/N and C/P were lesser in the N treatments compared to CK, whereas those of litter N/P was greater at the end of the experiment (Figure 4, $p < 0.05$). The effects of litter type, sampling time, and N addition level, and their interactions on the litter C/N, C/P, and N/P ratios were significant during the decomposition ($p < 0.01$, Table 2). Regardless of the N treatment, the C/P, C/N, and N/P ratios were significantly greater in twig litter than in leaf litter during two years of decomposition.

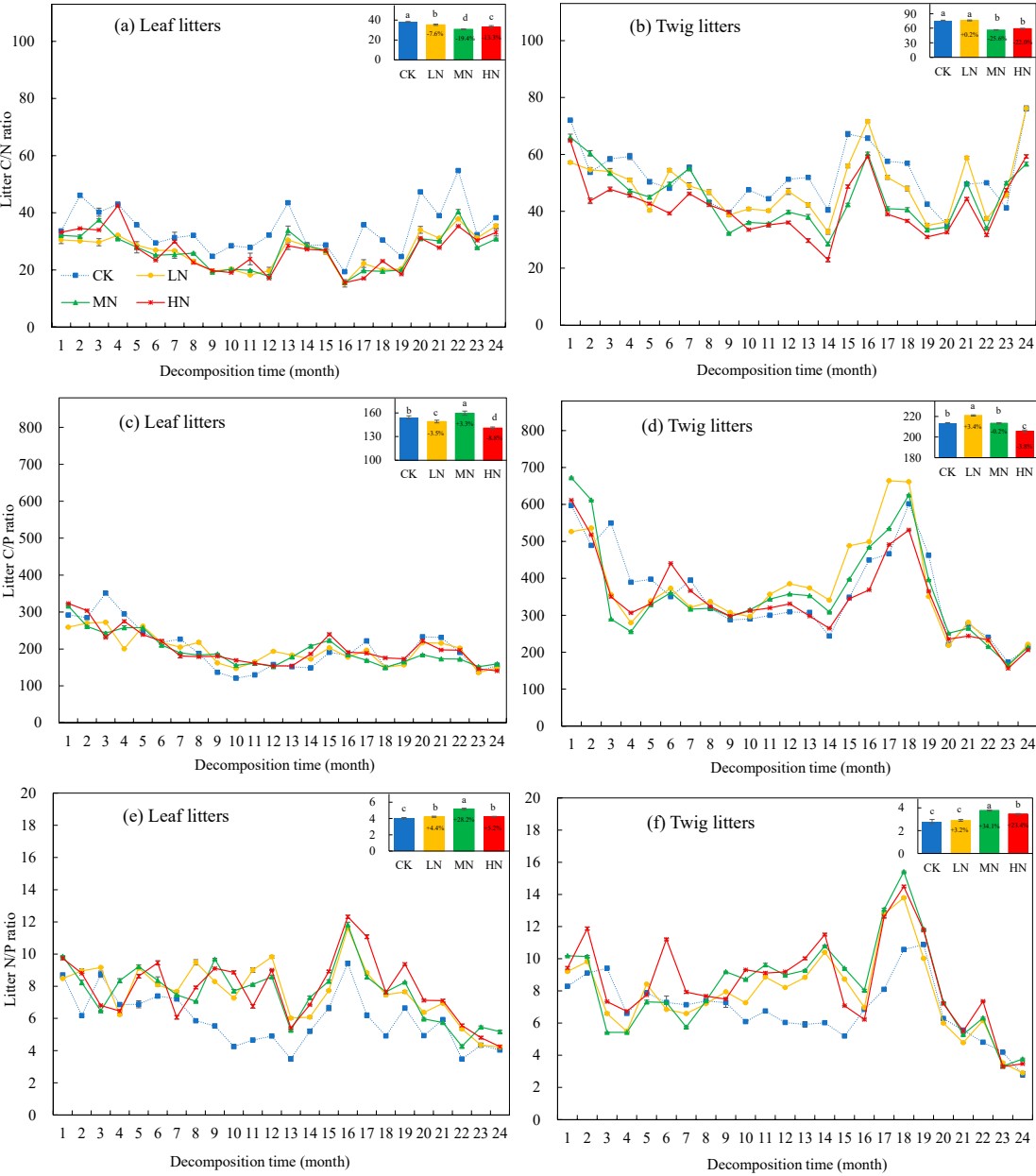

**Figure 4.** Remaining stoichiometry ratios in leaf and twig litters (% of the initial amount) during the decomposition process: C/N (**a**,**b**), C/P (**c**,**d**), and N/P (**e**,**f**). Based on mass, the litter stoichiometry ratios have been derived. Different histogram lowercase letters represent statistically significant differences between various treatments at the end of the experiment on the basis of three-way ANOVA ($p < 0.05$). Estimated values are the average of three plot replicates $\pm$ SE ($n = 3$).

*3.5. Linking Nutrient Releases with Lignin and Cellulose Degradation*

The remaining lignin in leaf litter quadratically correlated with the contents of C ($R^2 = 0.915$–$0.954$; $p < 0.01$; Figure 5a), N ($R^2 = 0.566$–$0.896$; $p < 0.01$; Figure 5c), and P ($R^2 = 0.518$–$0.744$; $p < 0.01$; Figure 5e). The remaining lignin in twig litter was also quadratically correlated with the remaining C ($R^2 = 0.712$–$0.823$; $p < 0.01$; Figure 5b) and N contents ($R^2 = 0.499$–$0.621$; $p < 0.01$; Figure 5d), whereas there was no significant correlation of the lignin contents with the remaining P (Figure 5f).

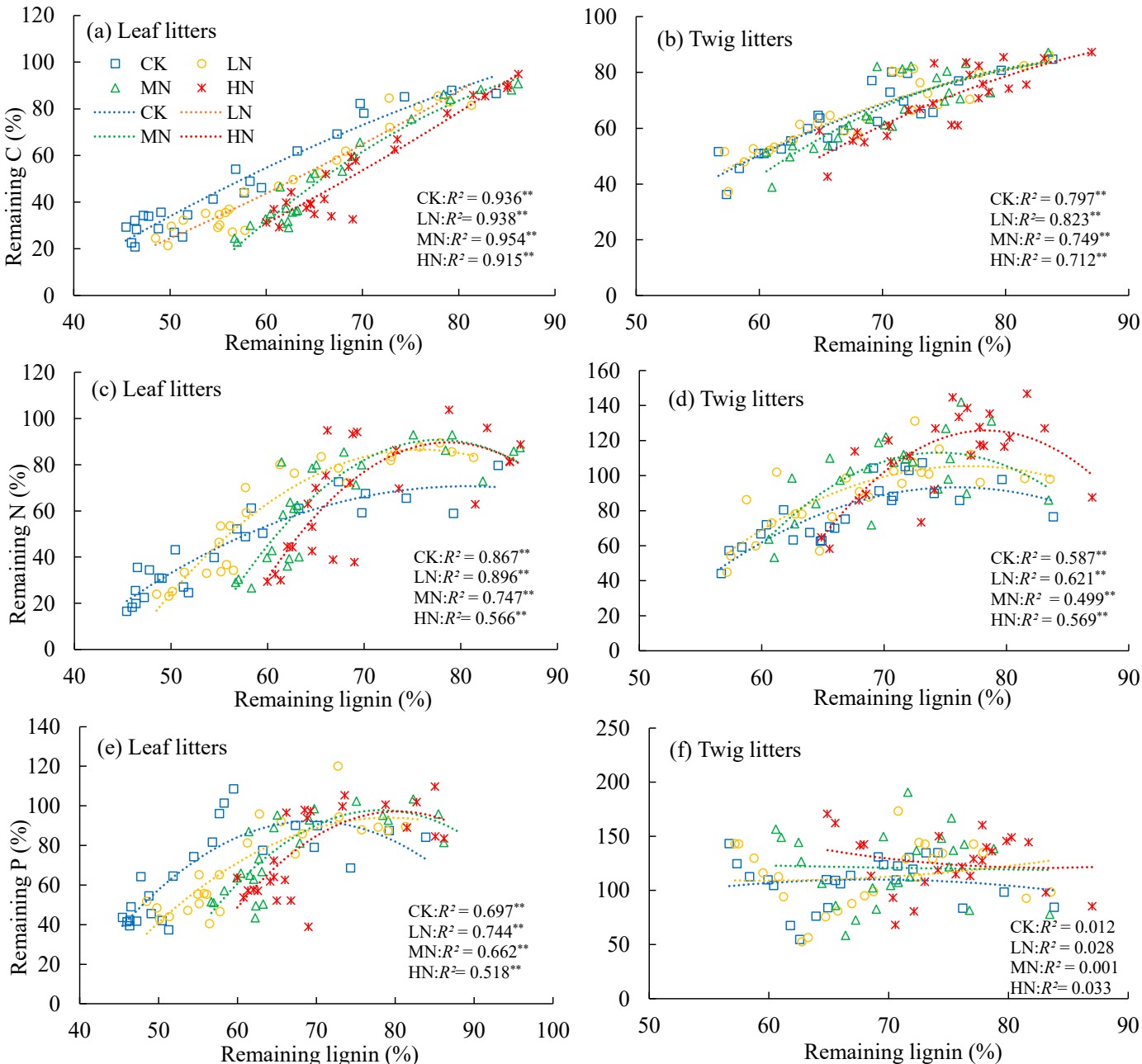

**Figure 5.** Statistical correlation between the remaining nutrients in leaf and twig litters (% of the initial amount) and remaining lignin (% of the initial amount). (**a,c,e**) The correlation between C, N, and P contents remaining and the remaining lignin of the leaf litter; (**b,d,f**) show the correlation between the amount of C, N, and P remaining and the remaining lignin in twig litter. ** Indicates that the determination coefficient is statistically significant at $p < 0.01$. At each sampling time, data were aggregated. The treatment abbreviations have been defined in Figure 1. $n = 576$ in leaf litters or twig litters.

The remaining cellulose in the leaf litter was correlated quadratically with the contents of C ($R^2$ = 0.951–0.977; $p < 0.01$; Figure 6a), N ($R^2$ = 0.624–0.882; $p < 0.01$; Figure 6c), and P ($R^2$ = 0.663–0.786; $p < 0.01$; Figure 6e). The cellulose remaining in twig litter was also correlated quadratically with the contents of C ($R^2$ = 0.868–0.920; $p < 0.01$; Figure 6b) and N ($R^2$ = 0.584–0.766; $p < 0.01$; Figure 6d). In contrast, there was no significant correlation observed between the cellulose and P contents remaining in twig litter (Figure 6f).

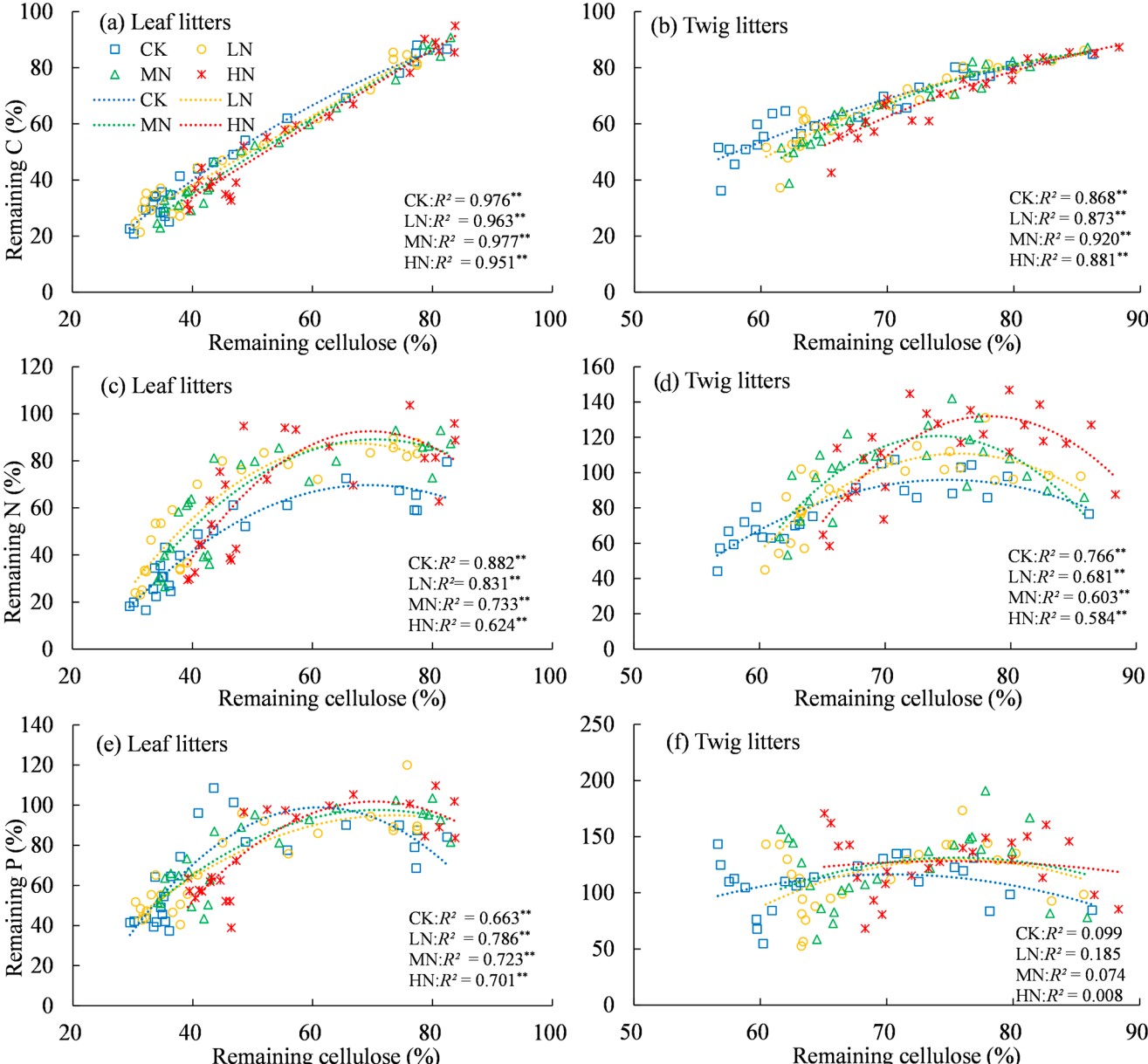

**Figure 6.** Statistical correlations between the nutrient remaining (% of the initial amount) and cellulose remaining (% of the initial amount) in the leaf and twig litters. (**a,c,e**) The correlation between the amount of C, N, and P remaining and remaining cellulose in leaf litter; (**b,d,f**) the correlation between the amount of C, N, and P remaining and remaining cellulose in twig litter. ** Indicates that the determination coefficient is statistically significant at $p < 0.01$. Each sampling time data were aggregated. The treatment abbreviations have been defined in Figure 1. $n$ = 576 in leaf litters or twig litters.

## 4. Discussion

### 4.1. Impacts of N Additions on Litter Decomposition

The current study indicated that the N additions significantly decreased the litter decomposition. The negative effects varied with N addition rates, which can validate our first hypothesis regarding the role of N addition in inhibiting the decay of leaf and twig litters in the forest area. Most prior researches on subtropical forests under different nitrogen deposition confirmed these results in a bamboo, natural evergreen broad-leaved, or Chinese fir forests [3,19,22,24,26]. This process can be explained by two underlying mechanisms. Among them, the first one is where the external N input can reduce the activity and community composition efficiency of soil microbial decomposers, thereby slowing the litter decomposition rate [22,33]. Previous studies conducted in subtropical forests have demonstrated that the N addition decreases the contents of C and N in the soil microbial biomass, as soil microbial biomass has a strong relationship with litter decomposition [22,34]. The N addition treatments may suppress the decomposition of litter in this study, even after providing more exogenous N for biological enrichment. The reason is that the high availability of N restrains the microbial access to N through organic matter [22,35]. On the other hand, a high level of N input consequently slows down litter decomposition due to the enrichment of soil N and exacerbation of soil C and P limitations, or even aggravating nutrient imbalance in the soil [36]. Tie et al. [3] demonstrated that the external addition of N exacerbated the limitations of soil C and P in subtropical evergreen broad-leaved forests, which may restrain litter decomposition. However, it was found in a subtropical forest of southwestern China that external N input simulated the decomposition of litter, which was found to be dominated by *Michelia wilsonii* [37], In general, when the initial litter C/N ratio is >55, additional inputs of N accelerate the leaf litter decomposition. In contrast, the N addition suppressed the litter decomposition in the *Camptotheca acuminata*—dominated forests, which may be attributed to the lower initial litter C/N ratio (<34) in this study. In particular, N was limiting in our evergreen broad-leaved forest (approximately 15 mg g$^{-1}$). Hence, the different responses of litter decomposition processes to N addition may be related to the activity of soil microorganisms and the initial substrate quality, as well as the nutrient dependence of ecosystems.

### 4.2. Impacts of N Addition on Nutrient Release

The nitrogen treatment resulted in the reduction in the C, N, and P release both in leaf and twig litter, except for P in twig litter. These findings partially support our first hypothesis that the addition of N shall lower the release of litter C, N, and P in the studied forest, which can be explained by two potential mechanisms. First, it is common knowledge that urea increases the external C and N nutrients in a direct manner, causing a slowdown in the release of these elements [38,39]. Furthermore, according to microbial nutrient mining theory (MNM), the release of nutrients is primarily determined by the stoichiometric requirements of the microbial decomposers [25]. When soil nutrients are plentiful, microbes do not have to degrade litter to obtain what they need, but when soil nutrients are deficient, bacteria can release litter nutrients to meet their needs for growth and reproduction [40,41]. Thus, microbes that digest litter immobilize nutrients from the environment to keep the stoichiometric equilibrium [41], meeting the requirement for microbes that cause C, N, and P to be released into ecosystems from litter [42]. The addition of urea can enhance the availability of nutrients in the ecosystem [43], which might have lowered MNM, thus attenuating the nutrient release. Nevertheless, other studies show the opposite result, that the N addition promoted the release of C and N in N-poor leaf litter [25,44]. At the end of the study period, the C, N, and P release from litters remained lowest in the HN treatment, which was consistent with an earlier study reporting that very high N additions suppressed the nutrient release from the litter [21]. This study unraveled that additions of N have a tendency to favor the P accumulation and help the immobilization of net P in litters during the whole experimental period [21,25]. This study unraveled that additions of N have a tendency to favor the P accumulation and help the immobilization of net P in litters during the whole experimental period [21,25]. On the one hand, this might be because

enzymatic processes mediated by phosphomono- and phosphodiesterase with N addition were inhibited as they lead to the decreasing release of phosphate from phosphomono- or phosphodiesters to the soil solution [45]; on the other hand, soils contained a large amount of P with the resin P pool and that microorganisms had taken up P from the resin P pool [45] and immobilized greater P in the litter to lessen the negative effect of N-induced litter P release to the soil, which results in an increased P demand after N additions [46]. Hence, the N addition might be working together with the N content of the initial litter to alter the nutrient release from the litter. In our study, a statistically significant positive relationship was found between the nutrient elements remaining and the cellulose content (quadratic), primarily because of the fact that cellulose accounts for approximately 40% of the mass of litter and is the primary substance lost during litter decomposition [47]. These results highlight that the cellulose remaining could help predict C, N, and P release from the litter.

The current study revealed that the impact of N addition on the release of N in litter varied across different components of litter. Regardless of the N regime, N release was faster in leaf litter than in twig litter, partly supporting our second hypothesis. The leaf litter tends to decompose more rapidly with a high decay rate, which might be related to the conspicuous differences in morphological components, physicochemical properties, initial chemistries, C/N ratio, and cellulose and lignin decomposition in leaf and twig litters [48,49]. Firstly, the twig litter is morphologically thicker and harder than the leaf litter. In terms of composition, the chemical components of twig litter majorly consist of lignin, cellulose, and phenols [10]. The leaf litter have an elevated content of compounds that are labile, facilitating the swift exploitation of microbial decomposers [29]. The high level of lignin and cellulose in twig litter will hinder their decomposition and even enrich lignin during the litter decomposition process. Secondly, previous studies have shown that in the decomposing litter, N commences releasing when the litter C/N ratio is <40 [50]. Our findings stated that the initial C/N ratio in leaf and twig litters was 32 and 65, respectively. Therefore, lower initial N content and higher C/N ratio, as well as higher lignin and cellulose residues in twig litter, could be one of the main reasons for their slower decomposition rate than the leaf litter [17]. In addition, Our outcomes demonstrated that the initial P content and the ratios of litter C/P and N/P might be critical factors that affect rates of decomposition. The leaf litter with higher P content decomposed faster than the twig litter with lower P content, probably because of the P limitation as a result of N addition, and the P effect may dominate the N effect [51]. A possible link between the stoichiometric constraint theory and the increase in litter P due to N deposition has been proposed [52]. In our study, we found that N additions tend to decrease the litter C/P and N/P ratios in the last stages of decomposition, which in turn reduces the rate of litter decomposition. More evidence is needed to confirm this hypothesis. Therefore, in southern Chinese subtropical evergreen broad-leaved forests, it is important to consider not only the reactions of decomposition of litter to the external nitrogen input but also the external phosphorus input.

### 4.3. Impacts of Lignin and Cellulose Degradation on Nutrient Release

Litter decomposition is strongly controlled by litter stoichiometry [53,54]. Changing stoichiometric ratios of nutrients at the time of decomposition can control successive decomposition and degradation processes of lignin and cellulose [36]. We provided evidence that the N addition slowed the lignin and cellulose degradation in litters (also higher in twig litter), which was consistent with a previous subtropical forest study [3,24]. This effect may be closely associated with the decline in the C/N ratio and the increase in the N/P ratio under conditions of N addition treatments. Firstly, lowering the C/N ratio led to an elevation in the number of bacteria in the microbial community waiting for such opportunities, which may benefit the accumulation of microbial necromass by increasing microbial C use efficiency. N addition treatments, thus, would probably reduce the fungi-to-bacteria ratio [55,56]. Furthermore, fungi have been reported to be more effective than bacteria

in degrading active organic matter to support high metabolic rates [55,57]. The present study's findings of a lower litter C/N ratio as a consequence of N treatments provided strong evidence that N treatments can reduce the fungi-to-bacteria ratio, thus preventing lignin and cellulose degradation [58]. Secondly, P is the primary factor that limits microbial activity in subtropical forests [59]. An elevated N/P ratio is deemed to intensify the limitation of phosphorus in microorganisms of nitrogen rich forests [3]. In the current study, the N treatments elevated the ratio of litter N/P at the time of the decomposition process, demonstrating that P limitation in microbes was exacerbated by N addition which reduced the cellulose and lignin degradation.

In all the treatments, the overall residual rate of lignin and cellulose decomposition decreased in litters, supporting our second hypothesis that the impact of N addition varied between two contrasting litter types due to different lignin and cellulose decomposition rates. The residual rate of lignin in litters was directly decreased during the whole stage, indicating that the content of lignin in litters decreased directly with an initial concentration of lignin greater than 300 mg g$^{-1}$ (approximately 400 mg g$^{-1}$ in our study). On the other hand, the content of lignin was found to increase at the early stage of decomposition before an absolute decrease [60]. The decrease in lignin residual rate may be due to the alternation in dry and wet climates in the study area, which accelerated the mechanical damage of litters (including the matrix metabolism and fragmentation of leaf and twig litters by soil animals) and further damaged the physical structure of lignin [61]. At the same time, the weather indirectly increased the specific surface area of litter and provided a larger niche for soil microorganisms. In addition, a warm environment and sufficient moisture content were conducive to the recovery of soil microbial abundance [62,63]. Therefore, the contribution of soil microorganisms to lignin decomposition was accelerated. In contrast, aromatic polymer reformation by the N addition might be a separate factor responsible for the attenuated degradation of lignin as a result of treatments. The N additions can enhance the number of polymers that are aromatic in nature by changing the fungi structure [58], which could be a possible underlying process that inhibits the lignin degradation under the N treatments conducted in the current study. We found that the residual rate of cellulose in litters also decreased with the decomposition time, which might be due to the decrease in soil microbial abundance and cellulase activity with litter decomposition [3]. In this study, the nutrient stoichiometry of two litter types can be taken as the indexes to represent the mechanisms of litter decomposition and nutrients release, while the soil microbial enzymes and community structure were also important factors in mediating the effects of N deposition on litter decomposition. Consequently, further work might be needed to pay particular attention to the soil microbial mechanisms when exploring the responses of leaf and twig decomposition to long-term N additions for expanding our understanding of nutrient cycling in the context of global changes [64].

## 5. Conclusions

The current study assessed the decomposition and the nutrient release from leaf and twig litters in a subtropical forest under N additions, where having a very high background nitrogen content. The high N additions significantly decreased the rate of litter decomposition (5.18%–7.78%) and the C, N, and P release (0.47%–32.63%). They also slowed down the lignin and cellulose degradation in litters (13.70%–25.90%), which was correlated with a reduced litter C/N ratio and an elevated litter N/P ratio. Moreover, a high effect of N addition on litter N release was observed for leaf litter than twig litter.

In particularly, N treatments elevated the litter N/P ratio and hence intensified litter P limitation during the decomposition processes.

The C, N, and P release from litters remained lowest in high N treatment, as observed at the study period's end. These findings underline the fact that in subtropical evergreen broad-leaved forests in southern China, the litter decomposition rate and the release of C, N, and P will lessen if N deposition rates continue to rise in this region.

**Author Contributions:** Y.S. and S.W. conceived and designed study; J.X. and C.H. performed the experiments; Y.S. and C.S. analyzed data; J.X. and Q.W. contributed reagents/methods/analysis tools; Y.S. and S.W. wrote the paper. All authors have read and agreed to the published version of the manuscript.

**Funding:** This research was supported by the Agricultural Joint Special Project of Yunnan Province (202301BD070001-059), First-class Discipline Construction Project of Yunnan Province (2022 No. 73), Natural Ecology Monitoring Network Project Operation Project of Yuxi Forest Ecological Station in Yunnan Province (2022-YN-13), Long-term Scientific Research Base of Yuxi Forest Ecosystem National in Yunnan Province (2020132550).

**Data Availability Statement:** The data are available on request from the corresponding author.

**Acknowledgments:** The authors thank the following people for their help in this research. Xiao-dong Li, Nai-mu Zhang, and Xu-ran Shang provided field assistance.

**Conflicts of Interest:** The authors declare no conflicts of interest.

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
