# Peer review of "Decomposition and Carbon and Nitrogen Releases of Twig and Leaf Litter Were Inhibited by Increased Level of Nitrogen Deposition in a Subtropical Evergreen Broad-Leaved Forest in Southwest China"

_forests, doi:10.3390/f15030492_

Round 1

Reviewer 1 Report

Comments and Suggestions for Authors

This study provides some interesting additional information about the effects of N additions on decomposition rates. However, the presentation needs improvement, and the authors may find my comments and suggestions helpful to improve the quality of this paper. Details of these comments and suggestions are reflected in the attached file. 

Comments on the Quality of English Language

The English is understandable. However, the presentation, structure, and grammar need further improvement. Probably, you may find my suggestion helpful. 

Author Response

Dear professor,

Thank you for offering us an opportunity to improve the quality of our submitted manuscript (forests-2869991). We appreciated the reviewers’ constructive and insightful comments very much. In this revision, we have addressed all these suggestions. We hope the revised manuscript has now met the publication standard of the journal.

We highlighted all the revision in red color, please see the word file attached.

Our point-to-point responses to the queries raised by the reviewers are listed.

Please check the word file.

Thank you very much!

Reviewer 2 Report

Comments and Suggestions for Authors

This study examined litter and twig decomposition and associated nutrient dynamics under different nitrogen loading levels in southern China, where nitrogen loading is expected to increase, and showed that increasing nitrogen loading level will hamper decomposition rates of litter and twig to change nutrient cycling in the ecosystems.

A simple hypothesis is presented, and experiments are conducted with a design to match the hypothesis, and the results obtained are discussed with statistical analysis.

The discussion is substantial and appropriately interprets the significance of the data. We see no problem with publishing this manuscript.

However, we would like to ask you to consider the following points.

Materials and Methods

Have you measured N deposition during the experimental period? This is useful to understand natural level of N load is in the control of the treatment.

The study site has a wet and a dry season. However, nitrogen source, soluble urea, is applied with a constant amount monthly, which is different from actual rainfall pattern at the site. I am wondering about the impact of this difference. I may be better to add some comments on this.

Relating to dry and wet seasonal pattern, time course changes in decomposition and nutrient release from litter and twigs may influenced by rainfall pattern. Please indicate the wet season and the dry season in the 24 months experiment in the figures. Or add an explanation in the text.

Line 142              Could you indicate the 5L water volume is equivalent to height (mm) of rainfall for the area of the experiment site?

Fig. 2     "Various lowercase histogram letters" is missing in the figure.

Fig. 3     In this figure, there is a valley in the 13th to 15th month of leaf litter decomposition pattern due to a decrease in nitrogen and phosphorus concentrations. There is also a depletion of phosphorus concentration in the 18th month of branch litter. This seems to affect the changes in C/N and N/P as well. Could you provide any reason or discussion as to the cause of this changes?

Fig. 5 and Fig. 6 What is the reason of selecting these fitting curves?  N and P seem to reach the ceilings. Could you make comments on this pattern?

Line 168 and 293 Where -> where

Line 206    the formula is wrong        = T -CK/CK -> = (T - CK)/CK

Line 235 - 237     I could not understand the numbers 5.4%, 23.3%, and 30.4%. Are these number come from table or figures?

Line 254 Fig. 4 -> Fig 3 ?

Please use large font size for the axes and legends of all figures. The letters are too small.

Regarding phosphate limitation in decomposition, there is a famous hypothesis by Prof. Peter Vitousek (Stanford University) that phosphorus is the limiting factor in nutrient cycling in tropical ecosystems. I do not know the experimental area in China, but I gently suggest that a related discussion may deepen your discussion.

 that is all.

Author Response

Dear professor,

Thank you for offering us an opportunity to improve the quality of our submitted manuscript (forests-2869991). We appreciated the reviewers’ constructive and insightful comments very much. In this revision, we have addressed all these suggestions. We hope the revised manuscript has now met the publication standard of the journal.

We highlighted all the revision in red color, please see the word file attached.

Our point-to-point responses to the queries raised by the reviewers are listed.

Please check the word file.

Thank you again!

Reviewer 3 Report

Comments and Suggestions for Authors

It is a very good and well written experimental article with interesting results contributing to understanding of the forest ecosystems functioning.

From theoretical point of view, this article demonstrates Shelford's law of tolerance in Ecology in relation to excessive life factors for organic debris decomposition in/on soil. It is a decrease of decomposition rate of forest litter at the strong increase of N input being higher of optimal values. The Authors give a comprehensive analysis and explanation of the mechanism of this phenomenon in the forest ecosystems under consideration.

Comments

Please expand the title by adding “...at increased level of nitrogen deposition in South-West China”. 

L. 68-69      “..twig litter play a crucial role in the cycling of nutrients in the forest ecosystem…” -This is not in all forests and it is not “crucial” but important

L. 76  (15 and 10 g N·m-2·yr-1, respectively) [19,20],..“    10 g N m-2 yr-1”  it is 100000 gN  ha-1 yr-1 or 100 kgN ha-1 yr-1:  it is a terribly high N deposition!  Is it true?   It is the same as a standard doze of N fertilization in agriculture. The maximal values were 60 kgN ha-1 yr-1 in 70th of 20 century in Europe.

L. 81 – 97      please edit the paragraph to clearly point out the aim of this study.

L. 112   “…a standard of approximately 20 years.”  what it means?       NB A short description of forest stand dendrometric parameters is necessary here.

L. 112– 114   Three lines on soils only!?  Soil of experimental site must be described in more detail with physico-chemical parameters (especially N and C:N) because the MS is devoted to SOIL processed of forest litter decomposition. The data on forest litter fall mass, input rate and minimally C:N ratio will be also necessary here.

L. 116     “2.2.1. The collection…”  Positions of sampling sites on the plots:  are they random or regular?

L.120   “…dried litter amounts were 6:4…”   is it a proportion of different litters? Edit the sentence please.

L. 135   …were determined….    were established… can be better

L. 137   Experiment:  20 and 25 g·N·m-2·y-1 additions   are high N inputs even at agricultural fertilization.  It is 200 and 250 kg N ha-1 every year plus 100 - 150 kg N ha-1 of atmospheric N deposition!  It is a specific case of strongly EXCESSIVE N inputs (the Authors mentioned this circumstance in the text, for example, line 77: “…extremely high N input levels…”) when the N concentration in soil can have neutral or even negative effect on decomposition and tree growth. And the results of this excellent experiment demonstrate the effect of Shelford’s law. Globally, forest trees and soils of subtropics and tropics have also a high rate of biological N fixation, therefore N is not a primary limiting factor there - in contrast with boreal forests.

L. 164  You did not determined “stoichiometry” in relation to soil processes here.

L. 165  “The stoichiometric rates of leaf and twig litters (C: N, C: P, and N: P)…“  Sorry (a) it is not “rates” - it is “ratios” and (b) why “stoichiometric”?  Stoichiometric ratios are related to organisms only but not to decomposing debris and soil.

L. 203 Fig 1      “SD” is not reflected on the figure

L. 245 Fig 2   “lowercase histogram letters” are absent on the figure

L. 262 Fig 3  There is a very strong variability of data with unexpected and significant N and P increase at second year. It is not accentuated and explained in Discussion.

L. 272  “3.4. Stoichiometric Ratios in the Litter”   again “Stoichiometric”

L. 405    “litter stoichiometry  = C:N, C:P, etc      “stoichiometry” here is not correct 

Author Response

(The authors gave the same response as above.)

Round 2

Reviewer 1 Report

Comments and Suggestions for Authors

I have no further comments on this paper. 

Reviewer 2 Report

Comments and Suggestions for Authors

I could not find your response in the manuscrip. Confirm this carefully.

Response: We read the relevant references of Prof. Peter Vitousek, which related to our study, he pointed out: “understanding nutrient availability across landscapes requires a spatially explicit assessment of the relative strength of depletion and enhancement”. We added it to the discussion section. The list of references is as follows. Thank you for your suggestion that could deepen our discussion. Porder, S.J., P.M. Vitousek, and G. Asner. Ground-based and remotely-sensed nutrient availability across a tropical landscape. Proceedings of the National Academy of Sciences 2005, 102:10909-10912 Asner, G.P., and P.M. Vitousek. Remote analysis of biological invasion and biogeochemical change. Proceedings of the National Academy of Sciences 2005, 102:4383-4386 

Author Response

Dear professor,

Thank you for offering us an opportunity to improve the quality of our submitted manuscript (forests-2869991). We appreciated the reviewers’ constructive and insightful comments very much. In this revision, we have addressed all these suggestions. We hope the revised manuscript has now met the publication standard of the journal.

We highlighted all the revision in red color, please see the word file attached.

Our point-to-point responses to the queries raised by the reviewers are listed.

Please check the word file.

Thank you very much for your comments and suggestions.

Your sincerely,

Dr. Yali Song
